# Measuring Health Literacy in Primary Healthcare: Adaptation and Validation of a French-Language Version of the Brief Health Literacy Screening among Patients with Chronic Conditions Seen in Primary Care

**DOI:** 10.3390/ijerph19137669

**Published:** 2022-06-23

**Authors:** Maud-Christine Chouinard, Mireille Lambert, Mélissa Lavoie, Sylvie D. Lambert, Émilie Hudon, Olivier Dumont-Samson, Catherine Hudon

**Affiliations:** 1Faculté des Sciences Infirmières, Université de Montréal, 2375, Chemin de la Côte-Ste-Catherine, Montréal, QC H3T 1A8, Canada; maud.christine.chouinard@umontreal.ca; 2Centre Intégré Universitaire de Santé et de Services Sociaux du Saguenay-Lac-Saint-Jean, 305 Saint-Vallier, Saguenay, QC G7H 5H6, Canada; mireille.lambert@usherbrooke.ca; 3Département des Sciences de la Santé, Université du Québec à Chicoutimi, 555, Boulevard de l’Université, Saguenay, QC G7H 2B1, Canada; melissa1_lavoie@uqac.ca (M.L.); emilie2_hudon@uqac.ca (É.H.); 4Ingram School of Nursing, St. Mary’s Research Centre, 680 Sherbrooke West, Suite 1800, Montréal, QC H3A 2M7, Canada; sylvie.lambert@mcgill.ca; 5Faculté de Médecine et des Sciences de la Santé, Université de Sherbrooke, 3001, 12e Avenue Nord, Sherbrooke, QC J1H 5N4, Canada; 6Département de Médecine de Famille et de Médecine d’Urgence, Université de Sherbrooke, 3001, 12e Avenue Nord, Sherbrooke, QC J1H 5N4, Canada; olivier.dumont-samson@usherbrooke.ca

**Keywords:** self-report, health literacy, validation study, primary care, chronic conditions

## Abstract

Background: The Brief Health Literacy Screening (BHLS) is a short self-report instrument developed to identify patients with inadequate health literacy. This study aimed to translate the BHLS into French Canadian (BHLS-FCv) and to evaluate its psychometric properties among patients with chronic conditions in primary care. Methods: The BHLS was translated into French using the Hawkins and Osborne’s method. Content validity was evaluated through cognitive interviews. A validation study of the BHLS-FCv was conducted in two primary care clinics in the province of Quebec (Canada) among adult patients with chronic conditions. Psychometric properties evaluated included: internal consistency (Cronbach’s alpha); test–retest reliability (intraclass correlation coefficient); and concurrent validity (Spearman’s correlations with the Health Literacy Questionnaire (HLQ)). Results: 178 participants completed the questionnaire at baseline and 47 completed the questionnaire two weeks later over the telephone. The average score was 13.3. Cronbach’s alpha for internal consistency was 0.77. The intraclass correlation coefficient for test–retest reliability was 0.69 (95% confidence interval: 0.45–0.83). Concurrent validity with Spearman’s correlation coefficient with three subscales of HLQ ranged from 0.28 to 0.58. Conclusions: The BHLS-FCv demonstrated acceptable psychometric properties and could be used in a population with chronic conditions in primary care.

## 1. Background

As a complex and multidimensional concept, health literacy has received growing attention from the scientific community during the last decades. Its scope has evolved over time, moving from focusing on individual skills, such as reading ability and numeracy skills, to include a more global representation in relation to political, social, cultural, and economic contexts [1]. From this perspective, the World Health Organization (WHO) describes health literacy as “the cognitive and social skills which determine the motivation and ability of individuals to gain access to, understand and use information in ways which promote and maintain good health” [2].

In many countries, low health literacy is highly prevalent. In North America, 40% to 60% of adults have inadequate health literacy [3,4]. Almost half of Europeans have limited health literacy [5,6]. Particularly vulnerable groups include older adults and those who report low income and less education. Another group at risk of low health literacy is those with chronic conditions, where 75% of them have limited health literacy abilities [7]. Inadequate health literacy directly contributes to negative outcomes, such as more hospitalizations, higher emergency department (ED) use [8,9], poorer health knowledge and medication adherence, poorer health status, and higher mortality [10,11,12].

Appropriate and valid measurements of health literacy are essential to identifying groups most at risk of low literacy to guide the design of interventions to improve health literacy and evaluate progress. It is especially important to assess health literacy in primary care practices where most of the patients have chronic conditions [13], and an increasing proportion of patients are managing and receiving information for more than one chronic condition [14,15]. To date, three measurement approaches of health literacy have been described: direct testing of individual abilities, and population-based proxy or self-report of abilities measures [16]. Among them, the most frequent approach has been direct testing of individual literacy skills such as reading ability and comprehension as well as numeracy skills. The most commonly used instruments include the National Assessment of Adult Literacy (NAAL) [17], the Rapid Estimate of Adult Literacy in Medicine (REALM) [18], the Test of Functional Health Literacy in Adults (TOFHLA) [19], the Newest Vital Sign (NVS) [20], the Health Literacy Skills Instrument (HLSI) [21], and the Functional, Communicative, and Critical Health Literacy scale (FCCHL) [22]. However, these instruments take time to administer and may require in-person testing by trained staff, both conditions less adapted to primary care settings. Most of them also involve more cognitive effort and could stigmatize or embarrass patients with low health literacy [23]. Accordingly [24], self-reported health literacy instruments, which are easy to apply in a clinical setting, have become increasingly popular in recent decades. Rapid and accurate screening tools for identifying patients with low health literacy could help guide interventions and improve care in primary care settings.

In a literature review to find brief instruments for identifying patients with limited literacy in clinical settings, the Brief Health Literacy Screening (BHLS) is recommended for rapid testing [25,26,27]. The BHLS is a brief self-report instrument (three items) for identifying patients with inadequate health literacy in research and clinical settings [28,29]. The instrument demonstrates adequate reliability with Cronbach’s alpha ranging from 0.71 to 0.79 as well as high validity, correlating significantly with the S-TOFHLA (*p* < 0.001) [26]. The BHLS has been translated into different languages [30,31,32,33], but no study has reported its psychometric properties among a French population in Canada. In 2020, Perrin et al. [34] assessed the content validity of the French translation of the BHLS [33] by conducting cognitive interviews with patients and providers in France. Although there are some differences between the French and French-Canadian translations, Perrin et al.’s conclusion remains relevant, the instrument is easy and quick to administer, but needs a psychometric evaluation and improvement to suit the needs of both patients and providers in current practice.

This study aimed to translate the BHLS into French Canadian (BHLS-FCv) and to evaluate its psychometric properties (internal consistency, test–retest reliability, and concurrent validity) among patients with chronic conditions seen in primary care.

## 2. Methods

### 2.1. Transcultural Adaptation

The transcultural adaptation process of the BHLS was conducted using the steps recommended by Hawkins and Osborne (Table 1) [35,36].

### 2.2. Validation Study

#### 2.2.1. Participants

Participants were recruited between 9 May and 26 May 2016, in the waiting rooms of two primary care clinics (i.e., Family Medicine Groups (FMG)). These FMGs were located in two cities in different regions (Saguenay-Lac-Saint-Jean and Estrie) of the province of Quebec, Canada. One FMG was in a rural area and included 7 practices. The other was in an urban area and included 21 practices. Participants were recruited while they waited for their healthcare provider appointment. Sampling was performed using the convenience method. The clinic’s receptionist handed out a flyer explaining the project to every patient. Participants had to (1) be a patient in the clinic, (2) be 18 years of age or older, (3) be a native French speaker, and (4) be afflicted with at least one chronic condition. Pregnant women and patients with unstable acute chronic conditions or unmanaged psychiatric or cognitive diseases were excluded.

#### 2.2.2. Measure

Two research assistants recruited patients while they were waiting for their appointment at the primary care clinic. Each patient was provided a description of the project that included a list of inclusion criteria. The research assistants explained the project to the patients and assessed their eligibility. Those who were eligible were asked for consent and to complete the questionnaire with the support of the research assistant (T1).

The questionnaire included:Sociodemographic questions on gender, marital status, education, occupation, and income;Disease Burden Morbidity Assessment (DBMA-21 items) used only to count the number of chronic conditions [38,39];Health Literacy Questionnaire (HLQ) used to measure health literacy that is frequently used in research settings [40,41]. The following subscales were selected to assess the concurrent validity because they covered the same dimensions as the BHLS questions: having sufficient information to manage one’s health (range from 1 to 4), ability to find good health information (range from 1 to 5), and understanding health information well enough to know what to do (range from 1 to 5). For the HLQ, a higher score means better health literacy;BHLS [29] contains three items addressing, namely, help with reading, confidence with forms, and problems with learning. It uses a five-point Likert-type scale that varies according to the questions: Questions 1 and 2: 1 “all of the time”; 2 “most of the time”; 3 “some of the time”; 4 “a little of the time”; 5 “none of the time”. Question 3: 5 “extremely”; 4 “quite a bit”; 3 “somewhat”; 2 “a little bit”; 1 “not at all”. The total score varied from 3 “highest problem” to 15 “no problem” related to health literacy.

Two weeks after the initial questionnaire, a research assistant contacted 47 participants by telephone to complete the questionnaire a second time (T2), but without the sociodemographic and DBMA sections. This timespan [42], as well as the number of participants [43], is considered adequate to evaluate test–retest reliability.

#### 2.2.3. Data Analysis

Data analysis was conducted using Statistical Package for the Social Sciences (SPSS) version 23.0. Some questionnaires were not fully completed by the patients in the clinic waiting room before they were called by their doctors for their appointment. These questionnaires were excluded from the analysis. The sociodemographic characteristics of the participants were described using mean and standard deviation (SD) for continuous variables (e.g., age and number of chronic conditions) and frequencies (n) and percentages (%) for categorical variables (e.g., occupation and income).

Measured psychometric properties of the BHLS-FCv include internal consistency, test–retest reliability, and concurrent validity. Cronbach’s alpha (α) was used to measure internal consistency: a score below 0.70 demonstrates less acceptable internal consistency; between 0.70 and 0.80, respectable internal consistency; between 0.80 and 0.90, very good internal consistency; and above 0.90, a need to reduce the number of items in the questionnaire [42]. The intraclass correlation coefficient (ICC) was used to measure test–retest reliability between T1 and T2: a score between 0.50 and 0.75 demonstrates low to moderate reliability; above 0.75 demonstrates good reliability [44]. Spearman’s correlation was used to measure concurrent validity as the Kolmogorov–Smirnov test showed a non-normal distribution. As correlation coefficients should be interpreted in the context of the posed scientific question [45] and based on previous studies that examined concurrent validity of the BHLS, we established that values greater than 0.25 can be considered as acceptable [46,47].

Based on the concurrent validity analysis using one sample Spearman correlation test, a minimum sample size of 128 participants was estimated based on the detection of a medium effect size (0.25) with α ≤ 0.05 and a power of 80%, using the power analysis feature of SPSS 23.0. A total of 178 participants were included in this study.

### 2.3. Ethics Approval

This study was approved by the ethics review boards of the Centre intégré universitaire de la santé et des services sociaux du Saguenay-Lac-Saint-Jean and the Centre intégré universitaire de santé et de services sociaux de l’Estrie-CHUS. All participants in the study completed and signed an informed consent form.

## 3. Results

### 3.1. Translation, Back Translation, and Evaluation Committee

After comparing all versions of the instrument (original English-language version, translated French-language version, and the version back-translated into English), a final French version of the BHLS was obtained. In Question 1, to include documents from all healthcare services, the word “hospital” was replaced by “health”. In Question 2, the notions of learning about medical conditions and understanding written materials were merged into one idea.

### 3.2. Pre-Test

Ten people with chronic conditions participated in the cognitive interviews. Many patients reported that they disliked the interrogative adverbs at the beginning of the questions (“How often”, “How confident”). The research team decided to replace this type of question with closed questions while keeping the five-point Likert-type scale (Table 2).

### 3.3. Validation Study

One hundred seventy-eight (*n* = 178) participants completed the questionnaire. The flow of patients during the study is presented in Figure 1. Of the 717 people approached to participate in the study, 484 were excluded. Of these, 313 did not meet the inclusion criteria, and 171 declined to participate. Of the remaining 233 participants, 178 completed the T1 questionnaire in full and were therefore included in the analysis. For the T2 questionnaire, 94 participants were contacted, but only 47 completed it (3 were excluded due to missing data in the BHLS, 35 could not be reached and 9 refused to participate).

Participants’ characteristics are presented in Table 3. For the 178 participants, the average age was 59 years. Most of the study participants were women (65.7%). Their education level ranged from 20% to 30% for each of the four categories. Half of the participants were retired, while one-third were employed. Only 13.5% of the participants reported having a household income of less than CAN 20,000 per year, while 46.8% reported having a household income of more than CAN 50,000 annually. More than half of the participants were married or lived with a partner (55.9%). No difference was observed between the characteristics of the total participant sample (*n* = 178) and the characteristics of participants who also completed the T2 questionnaire (*n* = 47).

BHLS-FCv validation results are presented in Table 4. The mean score of the BHLS-FCv was 13.3 (range = 3 to 15). Cronbach’s alpha score was 0.77. The ICC for test–retest reliability was 0.69 (95% CI: 0.45 to 0.83). For the concurrent validity, Spearman’s correlation between the three subscales of the HLQ and the BHLS-FCv varied from 0.28 to 0.58 (Table 5).

## 4. Discussion

The results of this study, the first evaluating psychometric properties of the translation of the BHLS into French Canadian, demonstrated acceptable psychometric properties of the instrument among patients with chronic conditions seen in primary care. Concerning concurrent validity, low to moderate correlation was found between the BHLS-FCv and HLQ subscales. Although data collection was conducted by research assistants rather than by health care providers in daily clinical practice, the literature has shown the validity of using the original version BHLS in clinical practice [26,27,28,29,30,31,32,33,34,35,36,37,38,39,40,41,42,43,44,45,46,47,48]. We can therefore assume that this is the case with the BHLS-FCv.

In validation studies conducted in 2004 and 2008, using the S-TOFHLA as the gold standard, Chew et al. concluded that the BHLS is effective in detecting inadequate health literacy, with the reported area under the receiver operating characteristic curve ranging from 0.66 to 0.87 [28,29]. Internal consistency of the BHLS was evaluated by Wallston [26] and yielded results similar to our study; Cronbach’s alpha for BHLS-RA (research assistant administered) was 0.79 among hospital patients (*N* = 498) and 0.71 among clinic patients (*N* = 295). No study evaluated test–retest reliability of the BHLS. Some studies validated translations of the BHLS in Arabic [30], Spanish [31], and Turkish [32], but none assessed internal consistency or test–retest reliability. Our study demonstrated moderate test–retest reliability. One explanation may be found in the difference between the methods of completion of the questionnaire used at T1 (self-administered in the waiting room) and at T2 (over the telephone with a research assistant).

Three studies evaluated the concurrent validity of the BHLS with the S-TOFHLA and reported reasonable correlations [26,27,28,29,30,31,32]. However, the BHLS is a self-report instrument, whereas the S-TOFHLA is a direct test of individual literacy skills. Therefore, it seemed more relevant to measure the concurrent validity of the BHLS with a self-reported measure such as the HLQ. This is the first study to measure the concurrent validity of the BHLS with HLQ subscales. Low to moderate correlations were found with the concept of literacy as measured by the three HLQ subscales. The HLQ was developed to measure the multiple dimensions of health literacy, such as the appraisal of health information, ability to find good quality health information, understanding health information well enough to know what to do, and actively managing health. On the other hand, the BHLS was developed to evaluate reading ability, comprehension, and numeracy skills, but was not intended to evaluate the way patients use health information to manage their health, which may explain the low to moderate correlations with HLQ subscales. In order to consider the complexity and multidimensionality of the concept of health literacy, the BHLS would benefit from the addition of items to measure, among other things, the ability to use acquired knowledge to manage one’s health or, as other researchers proposed, to measure, among other things, verbal health literacy [48].

Adaptations we made to the original BHLS instrument during the translation process related to the context of primary care are consistent with previous ones reported in the same context. As others have done previously [49,50], we changed items referring to hospital or medical aspects to ensure that they are more applicable in contexts outside hospital settings. We also observed that the interrogative adverbs at the beginning of the original three items could be difficult to answer, as Schulz et al. (2022) [51] had discovered before us, prompting us to avoid them.

### Strengths and Limitations of the Study

We applied rigorous linguistic and cultural adaptation methods to the questionnaire with the involvement of a patient and providers. The original author of the BHLS [28] was not involved in the translation process of the instrument, but she was aware of the study. The study took place in two different regions of Quebec (Canada), increasing the generalizability of the findings. Our results may not be applicable to patients on the lower end of the socioeconomic status scale, to less educated people, youth, or people without chronic conditions, as very few or none of the participants in our study fell into these categories. Indeed, people with these characteristics are likely among those who refused to participate (42.3%). Collecting data on “participants’ savings” or “perceived income adequacy” rather than on income by “brackets” could have supported results for people having higher income, but who also have significant financial obligations (e.g., debts, credit).

Future studies should consider concurrent validity with other self-report instruments measuring health literacy. Test–retest reliability could also be evaluated using the same methods of completion at T1 and T2. Sensitivity to change over time should also be considered further as well as validity in samples underrepresented in this study.

## 5. Conclusions

The BHLS-FCv demonstrated good internal consistency, moderate test–retest reliability, and low to moderate concurrent validity with three subscales of the HLQ. Policymakers, health managers, and health professionals in Canada can promote and use the BHLS-FCv in primary care settings to detect inadequate health literacy among patients with chronic conditions. This will allow for more targeted interventions to be put in place, resulting in better patient outcomes.

## Figures and Tables

**Figure 1 ijerph-19-07669-f001:**
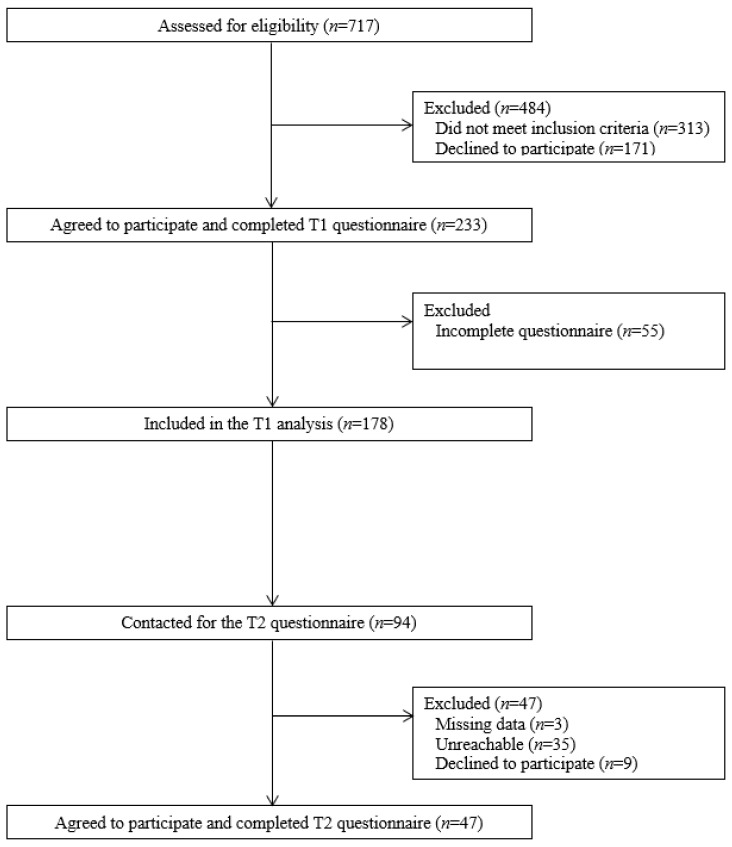
Flow chart of the participants.

**Table 1 ijerph-19-07669-t001:** Transcultural adaptation process steps of the BHLS.

Step	Description
Translation	The original English version was translated into French by a native French-speaking professional translator.
Back translation	The French version was translated back into English by a native English-speaking professional translator who did not see the original English version.
Translation evaluation committee	Both English versions, the original and the back-translated version, were compared by an expert group composed of two translators, one bilingual patient, and five members of the research team (MCC, CH, ML, ÉH, and EMC, four of them being healthcare providers). The aim was to identify any discrepancies between the versions to verify that the meaning was preserved in the translated French version.
Cognitive testing	A 30–45 min cognitive interview based on the Think Aloud Method [37] was administered to 10 patients with chronic conditions by a member of the research team (ÉH) to evaluate the validity of the translated French version of the questionnaire. Each patient was asked to read each question aloud and provide feedback on what they thought of it. The research team then examined problematic terms and modified them to improve patient understanding. Once patients no longer had difficulty completing the questionnaire, the final translated French version was prepared.
Validation of the French-language version	The final translated French version was completed by patients with chronic conditions seen in primary care. Internal consistency, test–retest reliability, and concurrent validity were evaluated. See details in the Validation study section.

**Table 2 ijerph-19-07669-t002:** English- and French-language versions of the BHLS.

BHLS	BHLS-FCv
How often do you have someone help you read hospital materials? - All of the time - Most of the time - Some of the time - A little of the time - None of the time	Avez-vous besoin d’aide pour lire des documents relatifs à la santé? - Toujours - La plupart du temps - Parfois - Rarement - Jamais
2. How often do you have problems learning about your medical condition because of difficulty understanding written information? - All of the time - Most of the time - Some of the time - A little of the time - None of the time	2. Avez-vous des difficultés à comprendre l’information écrite sur votre condition de santé? - Toujours - La plupart du temps - Parfois - Rarement - Jamais
3. How confident are you filling out medical forms by yourself? - Extremely - Quite a bit - Somewhat - A little bit - Not at all	3. Êtes-vous confiant pour remplir des formulaires par vous-même? - Extrêmement - Beaucoup - Moyennement - Un peu - Pas du tout

**Table 3 ijerph-19-07669-t003:** Participant characteristics.

Characteristics	*n* = 178	*n* = 47 *
Age (in years): mean (SD)	59 (15.8)	58.8 (15.3)
Number of conditions: mean (SD)	4.2 (2.3)	4.3 (2.0)
Female: *n* (%)	117 (65.7)	29 (61.7)
Education completed: *n* (%); 6 missing		
Less than high school	40 (23.3)	9 (19.1)
Completed high school	51 (29.7)	9 (19.1)
College or post-secondary institution	44 (25.6)	15 (31.9)
University	37 (21.5)	12 (25.5)
Occupation: *n* (%); 6 missing		
Employed	57 (33.1)	16 (34.0)
Unemployed	27 (15.7)	3 (6.4)
Retired	85 (49.4)	26 (55.3)
Other	3 (1.7)	1 (2.1)
Annual household income (in CAN$): *n* (%); 7 missing		
Less than CAN 20,000	23 (13.5)	3 (6.4)
CAN 20,000 to 49,999	68 (39.7)	22 (46.8)
CAN 50,000 or more	80 (46.8)	20 (42.6)
Marital status: *n* (%); 1 missing		
Married, living with a partner	99 (55.9)	24 (51.1)
Separated, divorced	28 (15.8)	10 (21.3)
Widowed	23 (13.0)	3 (6.4)
Single	27 (15.3)	10 (21.3)

SD: Standard deviation. * Subgroup who participated to test–retest.

**Table 4 ijerph-19-07669-t004:** Mean score and reliability of the three health literacy screening questions.

	Mean Score (Min–Max)	Internal Consistency, Cronbach’s Alpha	Test–Retest Reliability, ICC (95% CI)
T1 (*n* = 178)	13.3 (3–15)	0.77	0.69 (0.45–0.83)
T2 (*n* = 47)	13.1 (6–15)	0.79	

ICC: Intraclass correlation coefficient; CI: Confidence interval.

**Table 5 ijerph-19-07669-t005:** Concurrent validity of the three health literacy screening questions (*n* = 178).

HLQ Subscales	Concurrent Validity, Spearman’s Correlation
Having sufficient information to manage my health	0.28 *
Ability to find good health information	0.42 *
Understanding health information well enough to know what to do	0.58 *

* *p* ≤ 0.01. HLQ: Health Literacy Questionnaire.

## Data Availability

The data presented in this study are available on request from the corresponding author (C.H.). The data are not publicly available due to the Canadian regulatory framework (https://ethics.gc.ca/eng/policy-politique_tcps2-eptc2_2018.html, accessed on 5 March 2022).

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
