# Peer review of "Measuring Health Literacy in Primary Healthcare: Adaptation and Validation of a French-Language Version of the Brief Health Literacy Screening among Patients with Chronic Conditions Seen in Primary Care"

_ijerph, 2022, doi:10.3390/ijerph19137669_

Round 1
Reviewer 1 Report
Chouinard et al BHLS translation and validation
This paper reports the results of the translation and validation process of the BHLS (brief health literacy screening) in a French speaking primary care patients population with chronic disease. I thank the editor for the opportunity to review this interesting and well written paper. This study treat a topic of relevance and importance for the journal. The findings provide an important contribution to the literature since few literacy measurement tools are currently available and rigorously validated in french language.
Background :
- Majority of references are quite old, although some remain relevent because they refer to definitions, others refering to level of literacy of association between literacy and health status might be updated (notably p2 lines 46 to 52).
- p2 lines 54-57 & 65-68: it is not clear whether the authors are referring to the measurement of health literacy in everyday clinical practice or in a research context. It might be interesting to better distinguish between the two approaches and briefly add information about the implementation of literacy assessment in daily clinical practice in primary care in the present study (is it usual to assess the literacy level of patients in primary care and would the BHLS be suitable for the health care provider in this context).
- The Functional, Communicative and Critical Health Literacy scale has also been validated in french, this scla eis not mentioned in the introduction while it is also a short scale that might be used in clinical practice (Ousseine YM, Rouquette A, Bouhnik AD, Rigal L, Ringa V, Smith A', Mancini J. Validation of the French version of the Functional, Communicative and Critical Health Literacy scale (FCCHL). J Patient Rep Outcomes. 2017;2(1):3.)
Methods
The method used by the author is clear, rigorus and appropriate to the objective. However some parts of the study need further details.
- Translation : what is the justification for the interview sample size ? did you use a standardized approach (or interview grid) for the cognitive interviews ? if so please provide details about it.
- Please more explicitely detail the rule or recommandation on which you based your choice to include 120 participants (Polit’s recommandation is not specific to this design of questionnaire validation and other guidelines exists to define the required sample size – 10 participants by item for instance)
- It may be usefull to present the range of values of the HLQ dimensions scores min/max and direction of variation regarding the level of literacy.
- The scoring of the BHLS used in this study is not totally clear, if I correctly understand, the BHLS is interpreted in the paper with a total score being 0 to 4? It is usually scored with the sum of the items of each question with a 0-to-12 or 3-to-15 total score, with higher scores indicating higher health literacy. Can the authors clarify how they analyse the BHLS ? This impacts the interpretation of results regarding the mean score that is presented in table 4 and the negative correlations for the concurrent validity in table 5.
- A threshold of 9/15 to descriminate between adequate and low health literacy has also been proposed in the literature (see Willens DE, Kripalani S, Schildcrout JS, Cawthon C, Wallston K, Mion LC, Davis C, Danciu I, Rothman RL, Roumie CL. Association of brief health literacy screening and blood pressure in primary care. J Health Commun. 2013;18 (suppl 1):129–142. for instance), did the author consider using a threshold to classify patients into adequate or low literacy patients?
- As for other coefficients, cut-offs have been proposed to assess the level of correlation for the Spearman correlation coefficient, did the author used some cut-offs to interpret the Spearman correlation coefficients?
Results
- A high number of patients refused to participate (171). Did the authors collected reasons or characteristics of refusals to participate ? Is it possible that participants who refused had a lower helath literacy level and that this may have induce a selection bias in the study ? This may be more emphasized in the limtiation section of the discussion.
- The authors mentioned in the methods that they collected the DBMA but results are not presented in the paper. This is of importance for describing the population and to understand how participants were impacted by chronic conditions. Results of the HLQ dimensions should as well be presented in the results.
- tables 4 and 5 see previous remarks on BHLS scoring
Discussion
- This section does not discuss the influence of the primary care setting of the study while the title and version of the questionnaires are clearly oriented towards the primary care setting, discussion does not adress this point. Different versions of the questionnaire have been developped in other settings or population, it may be interesting to discuss the need for specific questionnaires.
Reviewer 2 Report
Results
Line 171-173: please provide the most important data from table 3 and figure 1
Line 171-173: the sentences concerning figure 1 and table 3 should be drafted like the sentence in line 173: "(...) is presented in (...)"
Line 173: it is written: "BHLS-Fv analysis is presented in Table 3." Please indicate these values.
Line 179: Table 3 presents data for n = 178, why did the authors not provide the results for n = 50? This remark applies to the entire results section. Results for n = 50 are nowhere to be found. I suggest you compare the results for n = 178 and n = 20.
Table 3: Annual household income is presented in the form of an x to x salary bracket. This pattern tells us nothing about the household economy. It is suggested that in this type of questions, ask, for example, whether the money is enough for each month, whether you can save money each month, etc. Sometimes people who earn a lot live modestly (because they have financial obligations, e.g. credit, debts), and low earners may run a higher level household because they have no credit, etc. Please refer to this in the discussion.
Tables 4 and 5: for which sample the data are presented: T1 (n = 178) or T2 (n = 50)
Line 190: (...) the translation of the BHLS into French (...) or French Canadian (see line 84)
Line 201-204: this is not shown in the results section. Please complete the results with a comparison between T1 and T2
References: Please complete with publications from 2022 and 2021
Reviewer 3 Report
This article translated the Brief Health Literacy Screening (BHLS) into French and issued questionnaires to French-Canadian patients with chronic diseases. The results demonstrated good internal consistency, moderate test-retest reliability and low to moderate concurrent validity with three subscales of the HLQ. The language of the article is concise and the experimental process is described in detail. In addition, the authors also recognized the shortcomings and limitations of the article, and proposed future research directions and plans. I agree to this article being published on ijerph under minor change. Hence, I have some small suggestions for the article.
1. It would be better to keep track of the time when the participant filled out the questionnaire in the appointment room. People who rushed to complete the questionnaire may not ruminate over their status and give an inaccurate result. This may be a reason why the test-retest reliability is not high enough.
2. The authors may appropriately add more historical literature to illustrate other methods of studying health literacy, which will help to highlight the contribution of this paper.
3. The Conclusion section of the article could tell us who will be interested in the results of this article and how the results of this article will help them.
